# TRANSFER ACTIVE LEARNING FOR GRAPH NEURAL NETWORKS

## ABSTRACT

Graph neural networks have been proved very effective for a variety of prediction tasks on graphs such as node classification. Generally, a large number of labeled data are required to train these networks. However, in reality it could be very expensive to obtain a large number of labeled data on large-scale graphs. In this paper, we studied active learning for graph neural networks, i.e., how to effectively label the nodes on a graph for training graph neural networks. We formulated the problem as a sequential decision process, which sequentially label informative nodes, and trained a policy network to maximize the performance of graph neural networks for a specific task. Moreover, we also studied how to learn a universal policy for labeling nodes on graphs with multiple training graphs and then transfer the learned policy to unseen graphs. Experimental results on both settings of a single graph and multiple training graphs (transfer learning setting) prove the effectiveness of our proposed approaches over many competitive baselines.

## 1 INTRODUCTION

Graphs encode the relations between different objects and are ubiquitous in real-world. Learning effective representations of graphs is critical to a variety of applications. Recently, graph neural networks (Kipf & Welling, 2016; Veličković et al., 2017) have been attracting growing attention, which iteratively update node representations by combining information of the nodes and their neighbors. They have been proved effective in many applications such as node classification (Kipf & Welling, 2016; Veličković et al., 2017) and link prediction (Zhang & Chen, 2018; Cao et al., 2018).

Despite the appealing performance, the training of graph neural networks typically requires massive labeled data (Yang et al., 2016). However, in many domains such as healthcare, a large number of labeled data may not be available, and it could be very expensive to obtain labeled data. Therefore, in this paper we focus on how to minimize human efforts in obtaining labeled data. Specifically, we study active learning for graph neural networks. In other words, our goal is to select informative nodes on graphs and query for their labels to train graph neural networks.

There are already some existing work that studied active learning for graphs (Gadde et al., 2014; Ji & Han, 2012; Cai et al., 2017), which generally used some optimization theory or proposed different kinds of heuristics (e.g., using the degree of nodes or the entropy of predicted label distributions) to select informative nodes for labeling. However, the nodes selected by these heuristics may not be optimal for a specific task. Moreover, these rules are usually domain specific, which are difficult to generalize to new graphs.

In this paper, we formulate active learning on graphs as a sequential decision process and aim to learn a policy network to sequentially pick the informative nodes for optimizing a specific task. Taking the node classification as an example, we define the *state* as the graph with the node information as whether the nodes are selected or not and their label distributions predicted by current graph neural networks. The *action* is picking the next node for labeling. After a node is picked, we retrain the graph neural network by adding the new picked node into the training set. The performance gain on the validation data set is defined as the *reward*. A policy network, which encodes the state with another graph neural network and predicts the next node to label, is trained to optimize the performance of graph neural network for the specific task.

The above approach allows us to learn a policy to select informative labeled nodes for training graph neural networks on a single graph. However, a more important question is whether the learned policy can be transferred to new unseen graphs. Therefore, we further study *transfer active learning for graph neural network*, which aims to learn a universal policy with multiple training graphs to generalize to new graphs. Given multiple training graphs, a straightforward solution is to learn a single policy across multiple training graphs. Such an approach works reasonable well in practice. However, in some cases, the optimal policy on different graphs may be different from each other. Therefore, we allow each graph to have a *graph-specific* policy, which is an instantiation of the universal policy on a different graph. During training, each graph-specific policy is trained to maximize the performance of its own task and meanwhile regularized by the universal policy, and the knowledge learned by different graph-specific policies is dynamically distilled into the universal policy by minimizing the KL divergence between the action distributions. With such a strategy, the universal policy can acquire knowledge from multiple graphs, and is therefore capable of transferring to new graphs.

We evaluate our proposed methods on the standard semi-supervised node classification task, in which the policy is trained and evaluated on a single graph, and the transfer active learning setting, in which the universal policy is trained with multiple graphs and evaluated on new graphs. Experimental results on both settings prove the effectiveness of our proposed approaches over many competitive baselines.

## 2 RELATED WORK

There are some existing work on active learning for graphs. Early studies (Guillory & Bilmes, 2009; Ji & Han, 2012; Gadde et al., 2014) typically formalized the task as an optimization problem. There are also works on semi-supervised classification on graphs with very few labeled data, such as Gallagher et al. (2008); Lin & Cohen (2010). Nevertheless, these methods are very computational expensive and cannot scale up to real-world graphs. Some recent studies (Cai et al., 2017) approached active learning by linearly combining several hand-crafted features including the uncertainty of the predictions, structure features such as centrality, and semantic features. More recently, Abel & Louzoun (2019) proposed to take the diversity of a node's neighbors in terms of label into consideration and choose the node to be labeled in a more regional perspective. However, the labeled nodes selected by these heuristics may not be optimal for the specific task while our policy network is trained to optimize the performance of the specific task. Moreover, these heuristics could be difficult to generalize to new unseen graphs.

Our work is also related to learning to learn strategies for active learning. Fang et al. (2017) studied using reinforcement learning for active learning on the task of Named Entity Recognition (NER), where they need to decide which sentence should be queried to label the Named Entity inside it. They formalized the sentence selection as a sequential decision process to sequentially select the sentences to be labeled. Similarly, Liu et al. (2018a) used imitation learning to train an active learning policy for NER tasks. Liu et al. (2018b) applied reinforcement learning algorithm to learn the policy for active learning on Neural Machine Translation tasks. Bachman et al. (2017) proposed to learn the policy network for active learning via meta-Learning. Different from these works, in this paper we focus on active learning for graphs, in which the data (the nodes) to be labeled are highly correlated, and hence the problem is more challenging. Moreover, our goal is learn a unified policy for active learning to transfer across different domains or graphs.

## 3 METHODOLOGY

### 3.1 PROBLEM DEFINITION

Graph neural networks have been proved effective in a variety of predictive tasks on graphs but require many labeled data for training. In this paper, we focus on how to minimize humans' efforts in obtaining labeled data on graphs (i.e. active learning) for training graph neural networks. Take the node classification as an example. We formulate the problem as a sequential decision process and aim to learn a policy to select the most informative nodes and further query for their labels for training graph neural networks. Formally, for a graph $G = (V_G, E_G, F_G, L_G)$, where $V_G, E_G, F_G$

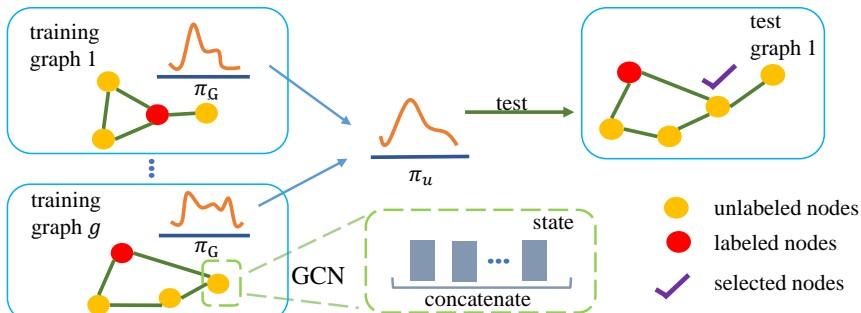

Figure 1: Overview of our model. The green box represents the components that forming the state. After that state is passed in to GCN to obtain a probability distribution $\pi_G$. Universal policy is distilled from $\pi_G$ and applied to test graphs.

and $L_G$ stand for nodes, edges, node features and node labels respectively, we aim at learning a policy $\pi_G$, which selects a sequence of nodes $V'_G \subseteq V_G$ from the training set with the size of $V'_G$ controlled by a given budget. We expect that the graph neural network trained with the labels of $V'_G$ has the optimal prediction performance on test sets of $G$.

Although active learning on single graphs is important, in practice we care more about whether the learned policy can transfer to unseen graphs. Therefore, we further propose to study *transfer active learning* for graph neural networks, where a universal policy is learned on multiple training graphs and evaluated on unseen test graphs. Formally, given a set of training graphs $S_{\text{train}}$, we seek to learn a universal policy $\pi_u$ for selecting informative labeled nodes on graphs, which is able to transfer to new unseen graphs in the test graph set $G \in S_{\text{test}}$.

We first introduce our approach for learning a policy network on a single graph and then introduce how to learn a universal policy with multiple training graphs.

## 3.2 ACTIVE LEARNING ON A SINGLE GRAPH

Recall that our goal is to learn a policy for selecting a set of nodes for annotation, such that a graph neural network trained with the labels of the selected nodes can have the optimal performance on test nodes. We formalize the problem as a sequential decision process (a.k.a., Markov decision process. We define the **state** as the current graph with the node attributes as whether they are picked or not and their label distributions predicted by the current graph neural network, and the **action** as picking a node for which we query the label. Once a node is picked, we will add this node into the training set and retrain the graph neural network, which will update the state. The performance gain on validation set is treated as the intermediate **reward**. The goal is to maximize the total sum of the reward, i.e., the final performance of the graph neural network. More specifically, for the state representation, we define the node features as follows:

- **Heuristic Features.** One effective criterion to active learning is the uncertainty of the prediction given by the training model, i.e., the entropy of the label distribution computed by the current graph neural network. Moreover, we also use the degree of a node, which is widely used in existing graph active learning methods (Cai et al., 2017). The details of heuristic features are presented in Appendix. Note that all the heuristics we use are transferable to other unseen graphs.
- **Structural Features.** Another observation is that nodes with similar structural roles should be similarly informative. Therefore, we also use some structural features. Specifically, we use the Struc2vec model (Figueiredo et al., 2017) to learn a latent representation for each node, which captures the local role of nodes. Struc2vec model allows joint training over different graphs, so once we learn the latent representations for all nodes of the graph that we are interested in, transfer learning can be done on this unified semantic space.
- **Historical Features.** Another important information is the historical information, i.e., which nodes are already annotated and which nodes are not. Motivated by that, we add an extra dimension in the feature vector of a node, where **1** means the node is already annotated, and **0** otherwise.

Moreover, we also average the struc2vec features of all previously annotated nodes to capture the historical information.

After getting the feature vector $\mathbf{x}_v^s$ defined above for each node $v$ at the state $s$, we pass them to a 2-layer graph convolutional networks to propagate the information of a node to its neighbors, and the final node representation, i.e., the state representation, is computed as $\mathbf{z}_v^s = \mathrm{GCN}(\mathbf{x}_v^s)$.

We solve the problem by training a policy network $\pi$, which defines a distribution over actions conditioned on states, i.e., the probability of selecting a node given the current state $\pi(a|s)$. We parameterize the policy network with a linear energy-based model as follows:

$$\pi(a = v|s) = \frac{1}{Z} \exp(\mathbf{w}^T \mathbf{z}_v^s), \tag{1}$$

where $\mathbf{w}$ is a weight vector, and $Z$ is the normalization term.

Our goal is to maximize the total sum of reward following the policy $\pi$, which can be characterized by the following objective function:

$$O_G(\pi) = \mathbb{E}_{s \sim d_\pi(s)} \mathbb{E}_{a \sim \pi(a|s)} [r_G(s, a)], \tag{2}$$

where $d_\pi(s) = \sum_{k=0} \gamma^k \mathrm{Pr}(s_0 \to s, k, \pi)$ is the stationary state distribution on graph $G$ induced by the policy $\pi$, with $\mathrm{Pr}(s_0 \to s, k, \pi)$ being the probability that the agent arrives at state $s$ in the $k$-th step starting from state $s_0$, and $\gamma$ being the decay factor. $r_G(s, a)$ is the reward function on that graph, which is computed as the performance gain on the validation set after taking action $a$ at $s$. Such an objective function can be effectively optimized by using the REINFORCE algorithm (Williams, 1992), where we sample some trajectories from the policy, and further update the parameters.

Next, we introduce how we learn a unified policy in the transfer learning setting.

### 3.3 TRANSFER ACTIVE LEARNING ON MULTIPLE GRAPHS

The above approach focuses on active learning on a single graph. In practice, a more important problem is to learn a universal policy from multiple training graphs, and further transfer the policy to unseen graphs. Towards this goal, we propose two effective approaches, i.e., a joint training approach and a distillation approach.

#### 3.3.1 JOINT TRAINING APPROACH

The joint training approach is quite straightforward. Basically, the universal policy $\pi_u$ is parameterized by using the policy network proposed in Section 3.2. We train the universal policy jointly on all the training graphs, resulting in the following objective function:

$$O_{\mathrm{Joint}}(\pi_u) = \sum_{G \in S_{\mathrm{train}}} O_G(\pi_u), \tag{3}$$

where $O_G$ is the objective function on each single graph defined in Eq. 2. Such an approach can be effectively optimized by stochastic gradient descent, where we combining the gradient computed from each single graph based on Eq. 3.

The problem of the joint training approach is that different graphs could have different optimal policies. Next, we introduce a more advanced approach to solve the problem.

#### 3.3.2 DISTILLATION APPROACH

Inspired by Teh et al. (2017), in the distillation approach, besides the universal policy $\pi_u$ we also introduce a graph-specific policy $\pi_G$ for each training graph $G \in S_{\mathrm{train}}$. Each graph-specific policy can be viewed as an instantiation of the universal policy on the corresponding graph, which aims at maximizing the performance (i.e., the total sum of the reward) for the specific task on that graph.

To coordinate the collaboration of different policies, we propose to minimize the following KL divergence between the action distributions computed by $\pi_u$ and each $\pi_G$:

$$\sum_{G \in S_{\mathrm{train}}} \mathbb{E}_{s \sim d_{\pi_g}(s)} \left[ \mathrm{KL}(\pi_G(a|s) || \pi_u(a|s)) \right]. \tag{4}$$

Intuitively, with this term, the universal policy serves as a bridge to connect different graph-specific policies, such that different graph-specific policies can exchange information during training. Meanwhile, by minimizing the KL divergence, the knowledge learned by different graph-specific graph policies will be effectively distilled into the universal query policy, which can be further applied to other graphs for active learning.

The overall objective function of the distillation approach is further obtained by combining the objective function on each single graph and the above regularization term, which can be formally written as follows:

$$O_{\text{Distill}}(\pi_u, \{\pi_G\}) = \sum_{G \in S_{\text{train}}} O_G(\pi_G) - c_{\text{KL}} \sum_{G \in S_{\text{train}}} [\mathbb{E}_{s \sim d_{\pi_G}(s)} \text{KL}(\pi_G(a|s), \pi_u(a|s))], \quad (5)$$

where $c_{\text{KL}}$ is a hyperparameter to control the weight of the KL term.

For each graph-specific policy $\pi_G$, it can be optimized by the REINFORCE algorithm. Formally, by taking a series of actions with $\pi_G$ on graph $G$, we obtain a trajectory $\{(s_{G,t}, a_{G,t}, r_{G,t})\}_{t=1}^T$. With the trajectory, the value of each state-action pair $(s_{G,t}, a_{G,t})$ can be further estimated as follows:

$$\hat{Q}_{\pi_G}(s_{G,t}, a_{G,t}) = \sum_{t'=t}^T \gamma^{t'-t} \{r_{G,t} + c_{\text{KL}} \log \pi_u(a_{G,t}|s_{G,t}) - c_{\text{KL}} \log \pi_G(a_{G,t}|s_{G,t})\}. \quad (6)$$

When computing $\hat{Q}_{\pi_G}(s_{G,t}, a_{G,t})$, besides the reward $r_{G,t}$ received at each step, two other terms are also incorporated to shape the reward function. For the first term $\log \pi_u(a_{G,t}|s_{G,t})$, it gives an action larger reward if the action receives large probability from the universal policy $\pi_u$. In this way, the universal policy provides extra guidance to regularize each graph-specific policy. For the second term $\log \pi_G(a_{G,t}|s_{G,t})$, it improves exploration by increasing the entropy of $\pi_G(a_{G,t}|s_{G,t})$, which is commonly used in the reinforcement learning literature Schulman et al. (2017).

With the estimated $Q$ value, we can further estimate the gradient for the parameters $\theta_G$ of $\pi_G$ as follows:

$$\nabla_{\theta_G} O_{\text{Distill}}(\pi_u, \{\pi_G\}) \simeq \frac{1}{T} \sum_{t=1}^T \hat{Q}_{\pi_G}(s_{G,t}, a_{G,t}) \nabla_{\theta_G} \log \pi_G(a_{G,t}|s_{G,t}). \quad (7)$$

Meanwhile, the universal policy can also be optimized through stochastic gradient descent. Specifically, we treat the trajectories generated by graph-specific policies as expert trajectories, and estimate the gradient for the parameters $\theta_u$ of the universal policy as follows:

$$\nabla_{\theta_u} O_{\text{Distill}}(\pi_u, \{\pi_G\}) \simeq \sum_{G \in S_{\text{train}}} \frac{1}{T} \sum_{t=1}^T \nabla_{\theta_u} \log \pi_u(a_{G,t}|s_{G,t}). \quad (8)$$

By using the trajectories from graph-specific policies as expert trajectories, we can effectively distill the knowledge into the universal policy, which can be further applied to unseen graphs.

The detailed optimization algorithm is summarized in Alg. 1.

---

**Algorithm 1** Pseudo code of our distillation-based approach.

---

1: **procedure** OPTIMIZEPOLICY $(\{G\})$
2:    **for** episode $i$ **do**
3:       **for** graph $G$ **do**
4:          initialize classification network and train with random initial seeds
5:          **for** step $t \leq N_{\text{budget}} - N_{\text{seed}}$ **do**
6:             feed $S_{G,t}$ to GNN and generate $\pi_G$ according to (1)
7:             action $a_{G,t} \leftarrow$ sample action from $\pi_G$
8:             reward $r_{G,t}$, new state $s_{G,t+1} \leftarrow$ update classification network using action
9:             update $\pi_G$, $\pi_u$ according to (7), (8), respectively
10:          **end for**
11:       **end for**
12:    **end for**
13: **end procedure**

---

Table 1: Dataset statistics

| | Graphs | Nodes | Edges | Edge density | Features | Classes |
|---|---|---|---|---|---|---|
| **Cora** | 1 | 2708 | 5278 | 0.0014 | 1433 | 7 |
| **Citseer** | 1 | 3327 | 4676 | 0.0008 | 3703 | 6 |
| **Pubmed** | 1 | 19718 | 44327 | 0.0002 | 500 | 3 |
| **Coauthor-cs** | 1 | 18333 | 81894 | 0.0005 | 6805 | 67 |
| **Cora-full** | 1 | 18707 | 62421 | 0.0004 | 8710 | 15 |
| **ppi** | 20 | 2245.3 | 61318.4 | 0.0248 | 50 | 121 (multilabel) |

## 4 EXPERIMENT

In this section, we empirically evaluate our proposed active learning approaches on several datasets. Specifically, we consider active learning on single graphs, transfer active learning on homologous graphs and transfer active learning on heterogeneous graphs.

### 4.1 DATASETS

- **Active learning on single graphs.** We utilize three benchmark datasets for node classification, i.e., Cora (Sen et al., 2008), Citeseer (Sen et al., 2008) and Pubmed (Namata et al., 2012).
- **Transfer active learning on homologous graphs.** For transfer active learning, we consider a set of homologous graphs from the PPI dataset (20 graphs) (Zitnik & Leskovec, 2017), where each graph is constructed based on the interactions between proteins. We randomly partition the 20 graphs into 4 groups. Within each group, two graphs are used for training, one graph for validation and the other two for test. For each training graph, we split all the nodes to train/validation/test sets with a ratio of 3:1:1. As there are too many node categories (121 categories in total), we randomly select 10 categories as our prediction target.
- **Transfer active learning on heterogeneous graphs.** To test whether our learned policy can transfer across graphs with different structures, we also consider a group of heterogeneous graphs. Specifically, we incorporate two extra graphs, i.e., Coauthor-CS and Cora-Full from Shchur et al. (2018), besides Cora, Citeseer and Pubmed. The Cora and Citeseer datasets are used for training, and we evaluate the learned policy on the Pubmed, Coauthor-CS and Cora-Full datasets.

The statistics of datasets are presented in Table 1, where for the PPI dataset, we report the average statistics of its 20 graphs.

### 4.2 COMPARED ALGORITHMS

We compare the following algorithms in experiment.

**(1) Random Policy**. At each state, the policy randomly selects a node and queries for its label. **(2) Entropy-based Policy**. At each state, we predict the label distribution for each node by using the current graph neural network, and further compute the entropy. The policy selects the node with the maximum entropy for annotation. **(3) Centrality-based policy**. This policy selects the node with the largest degree to query for the node label. **(4) AGE** (Cai et al., 2017). AGE is an active learning method for training graph neural networks. At each state, AGE measures the informativeness of each node by linearly combining several heuristics, including the entropy of the predicted label distribution, PageRank score and the cluster assignment. Then it selects the most informative node to annotate. To apply AGE to the transfer learning setting, we learn the optimal weight of different heuristics on the training graphs, and further transfer the weight to unseen graphs. **(5) DAG-Single**. Our proposed approach to active learning on single graphs. **(6) DAG-Joint**. Our joint training approach to transfer active learning for graph neural networks. **(7) DAG-Distill**. Our distillation-based approach to transfer active learning, where a universal and a set of graph-specific policies are learned in a collaborative way.

### 4.3 PARAMETER SETTINGS

All the compared methods aim to select a set of nodes, for which we query the node labels. Then a prediction graph neural network is trained by using the selected nodes, and the reward function is defined according to the performance of the graph neural network. In our experiments, we use a two-

Table 2: Active learning on single graphs

| Method | Cora | | Citeseer | | Pubmed | |
|---|---|---|---|---|---|---|
| | Micro-f1 | Macro-f1 | Micro-f1 | Macro-f1 | Micro-f1 | Macro-f1 |
| Random | $66.5 \pm 4.5$ | $58.2 \pm 8.0$ | $60.7 \pm 5.9$ | $53.0 \pm 6.5$ | $68.1 \pm 5.6$ | $66.5 \pm 7.0$ |
| Entropy | $68.0 \pm 3.6$ | $62.3 \pm 5.8$ | $62.4 \pm 4.0$ | $56.2 \pm 3.7$ | $69.8 \pm 4.4$ | $68.5 \pm 5.6$ |
| Centrality | $51.7 \pm 9.2$ | $36.8 \pm 13.4$ | $47.2 \pm 11.7$ | $38.3 \pm 12.8$ | $69.0 \pm 4.6$ | $67.3 \pm 5.9$ |
| AGE | $71.8 \pm 3.8$ | $\mathbf{67.4 \pm 5.5}$ | $68.9 \pm 2.5$ | $61.8 \pm 2.5$ | $74.7 \pm 4.3$ | $72.8 \pm 6.0$ |
| DAG-Single | $\mathbf{73.2 \pm 2.2}$ | $66.9 \pm 4.0$ | $\mathbf{71.1 \pm 2.0}$ | $\mathbf{62.4 \pm 2.7}$ | $\mathbf{79.1 \pm 2.3}$ | $\mathbf{78.0 \pm 2.1}$ |

Table 3: Transfer active learning on homologous graphs (PPI dataset)

| Method | 0-4 | | 5-9 | | 10-14 | | 15-19 | |
|---|---|---|---|---|---|---|---|---|
| | Micro-f1 | Macro-f1 | Micro-f1 | Macro-f1 | Micro-f1 | Macro-f1 | Micro-f1 | Macro-f1 |
| Random | $51.2 \pm 3.4$ | $31.0 \pm 6.0$ | $50.3 \pm 5.8$ | $41.6 \pm 7.0$ | $43.3 \pm 6.4$ | $32.1 \pm 7.8$ | $20.6 \pm 7.2$ | $15.4 \pm 5.5$ |
| Entropy | $47.5 \pm 3.7$ | $22.2 \pm 6.5$ | $40.0 \pm 9.8$ | $28.8 \pm 9.1$ | $34.4 \pm 7.7$ | $21.7 \pm 7.9$ | $13.6 \pm 7.5$ | $10.1 \pm 5.3$ |
| Centrality | $49.5 \pm 4.4$ | $26.2 \pm 8.2$ | $46.1 \pm 10.0$ | $35.8 \pm 11.3$ | $39.6 \pm 7.2$ | $25.2 \pm 11.0$ | $21.1 \pm 7.4$ | $13.8 \pm 7.2$ |
| AGE | $53.0 \pm 1.3$ | $41.4 \pm 2.3$ | $55.6 \pm 2.0$ | $51.2 \pm 2.5$ | $48.7 \pm 2.3$ | $42.6 \pm 3.0$ | $32.0 \pm 2.5$ | $28.0 \pm 3.5$ |
| DAG-Joint | $53.1 \pm 1.0$ | $42.0 \pm 1.3$ | $56.6 \pm 1.5$ | $52.9 \pm 1.8$ | $49.4 \pm 1.3$ | $43.5 \pm 1.8$ | $\mathbf{34.5 \pm 1.6}$ | $\mathbf{31.6 \pm 2.1}$ |
| DAG-Distill | $\mathbf{53.2 \pm 1.0}$ | $\mathbf{42.1 \pm 1.4}$ | $\mathbf{56.7 \pm 1.4}$ | $\mathbf{53.1 \pm 1.6}$ | $\mathbf{49.7 \pm 1.2}$ | $\mathbf{43.6 \pm 1.6}$ | $34.2 \pm 1.7$ | $31.3 \pm 2.0$ |

layer graph convolutional network (Kipf & Welling, 2016) as the prediction graph neural network, which is optimized by the Adam optimizer (Kingma & Ba, 2014) with a weight decay of 0.0005.

**Active learning on single graphs**. On single graphs, we use SGD to optimize our policy network with a learning rate as 0.01. The hidden dimension of the policy network is set to 128. The total budget (i.e., the total number of nodes to select) is set as $5 \times N_{class}$ by default, with $N_{class}$ being the number of node classes on each dataset.

**Transfer active learning**. On homologous graphs, the budget for active learning is set to 20. The weight of the KL term for the distillation approach is set to 0.0001. On heterogeneous graphs, we set the budget for active learning to $5 \times N_{class}$ by default, and the weight of the KL term is 0.0001.

We do 50 experiments for each setting and report the mean and standard error of micro-f1 and macro-f1. Note that because AGE is much slower than our algorithms, for Cora-full dataset, we only report the mean and standard error of 10 experiments when using AGE as the strategy.

### 4.4 COMPARISON WITH BASELINE ALGORITHMS

In this section, we compare our approach DAG against the baseline algorithms.

**Active learning on single graphs**. Table 2 presents the result of active learning on single graphs. Compared with the random policy and policies using simple heuristics such as entropy and centrality, our approach significantly outperforms them, since we consider more useful features. The AGE method also explores several features, but our approach still achieves better results especially on large graphs (Pubmed). This is because our approach parameterizes the policy network by using a deep architecture, which has better capacity than AGE, as AGE simply combines features in a linear way. Moreover, DAG is trained by using a reinforcement learning framework, which can do exploration more effectively.

**Transfer active learning**. Table 3 and table 4 present the results of transfer active learning. We see that both of our proposed approaches (DAG-Joint and DAG-Distill) significantly outperform all the baseline methods, showing that they can indeed learn an effective universal policy to transfer to the unseen graphs. Comparing DAG-Joint and DAG-Distill, we see that the simple joint training approach DAG-Joint achieves close results to DAG-Distill on the PPI dataset. This is because the graphs in the PPI dataset are homologous, which implies that different graphs have similar optimal policies, and therefore the policy learned by DAG-Joint is reasonably effective. On heterogeneous graphs, DAG-Distill achieves relatively better results.

### 4.5 PERFORMANCE UNDER DIFFERENT BUDGETS

The previous results have proved the effectiveness of our proposed approaches on single graphs and in the transfer learning settings, where a fixed number of budget (i.e., the total number of nodes to

Table 4: Transfer active learning on heterogeneous graphs

| Method | Pubmed | | Coauthor-cs | | Cora-full | |
|---|---|---|---|---|---|---|
| | Micro-f1 | Macro-f1 | Micro-f1 | Macro-f1 | Micro-f1 | Macro-f1 |
| Random | $68.1 \pm 5.6$ | $66.5 \pm 7.0$ | $85.0 \pm 3.0$ | $71.4 \pm 6.2$ | $46.3 \pm 1.8$ | $33.8 \pm 1.9$ |
| Entropy | $69.8 \pm 4.4$ | $68.5 \pm 5.6$ | $82.2 \pm 4.5$ | $77.7 \pm 5.9$ | $47.1 \pm 1.6$ | $33.1 \pm 1.8$ |
| Centrality | $69.0 \pm 4.6$ | $67.3 \pm 5.9$ | $68.4 \pm 12.9$ | $46.3 \pm 17.8$ | $31.4 \pm 9.1$ | $18.0 \pm 9.0$ |
| AGE | $74.7 \pm 4.3$ | $72.8 \pm 6.0$ | $88.1 \pm 1.7$ | $80.6 \pm 5.7$ | $49.1 \pm 1.4$ | $37.1 \pm 1.7$ |
| DAG-Joint | $75.6 \pm 3.5$ | $71.6 \pm 6.7$ | $88.9 \pm 1.3$ | $83.9 \pm 3.3$ | $\mathbf{51.4 \pm 1.2}$ | $\mathbf{39.8 \pm 1.3}$ |
| DAG-Distill | $\mathbf{76.4 \pm 3.1}$ | $\mathbf{73.3 \pm 5.5}$ | $\mathbf{89.3 \pm 1.1}$ | $\mathbf{85.6 \pm 2.6}$ | $49.8 \pm 1.3$ | $38.1 \pm 1.4$ |

query) is considered. Next, we study the performance of the compared algorithms under different budgets, where two cases are considered. In the first case, we train our policy on Cora and Citeseer with a budget in $\{2, 5, 10, 20\}$, and then we evaluate the learned policy on Pubmed by using the same budget. In the second case, we train the policy on 4 graphs of the PPI datasets with a budget in $\{1, 2, 5, 10\}$. Afterwards, the learned policy is applied to other graphs under the same budget.

The results are presented in Figure 2. We see that our proposed approach outperforms the strongest baseline algorithm AGE in most cases. Moreover, when the budget is small, DAG achieves relatively large improvement over AGE, which demonstrates that the policy trained by our approach can indeed learn to select the most informative nodes in a graph. In addition, one goal of active learning is to reduce the annotation efforts by achieving good performance with fewer annotations. In the first case, we see that the graph neural network trained by our approach achieves a micro-f1 score of 76% with 15 annotations, whereas AGE achieves the same micro-f1 score with 30 annotations. Therefore, our proposed approach can indeed help reduce the annotation efforts by learning an effective universal policy on several training graphs.

## 4.6 CASE STUDY AND ABLATION STUDY

We also conduct some case study to show what are the nodes selected by our trained policy network and an ablation study to show the importance of different types of node features for representing the states. The results are available at the appendix (Section A.1 and A.2).

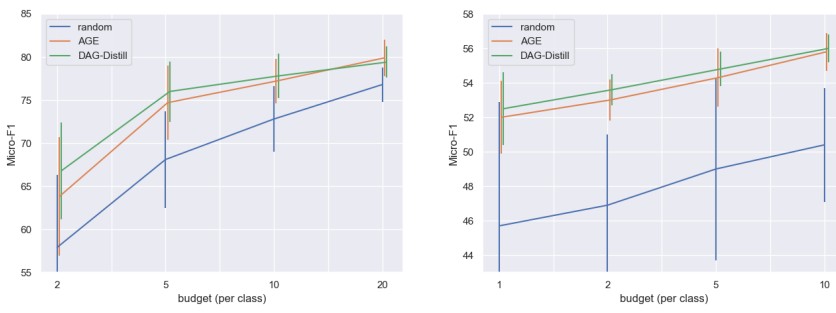

Figure 2: Performance of DAG-Distill under different budgets. Vertical lines represent the standard error of experiments

## 5 CONCLUSION

This paper studies active learning for graph neural networks, where two different settings are considered, i.e., active learning on single graphs and transfer active learning on multiple graphs. We formalize the problem as a sequential decision making process, and propose a policy gradient method on single graphs. For transfer reinforcement learning, a joint training approach and a distillation-based approach are proposed. Experimental results prove the effectiveness of our approaches. In the future, we plan to use our approach in larger datasets, where a large number of graphs are available.

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

# A APPENDIX

## A.1 CASE STUDY

To demonstrate the policy learned by our algorithm is indeed reasonable, we show the sequence of nodes selected by our algorithm. Taking the performance of DAG-Distill policy $\pi_u$ on Cora dataset as an example, for each step of node selection, we draw the subgraph formed by the selected nodes and their 2-hop neighbors. The predicted classes are represented in different colors and the probabilities of the predictions are represented by the transparency of the color (the higher the transparency is, the lower the probability is). Further more, the nodes of the largest size are selected in the corresponding step and the medium sized nodes were select previously. At the initial state of the node selection, the policy tends to choose the nodes in the center of region with less diversity of labels (e.g. step 1, step 3, step 5). In the middle stage, the policy prefers the nodes that serve as a bridge of two densely connected components (e.g. step 23, step 25). In the latter stage, the policy tends to choose the node with great diversity of neighbor (e.g. step 26, step 28).

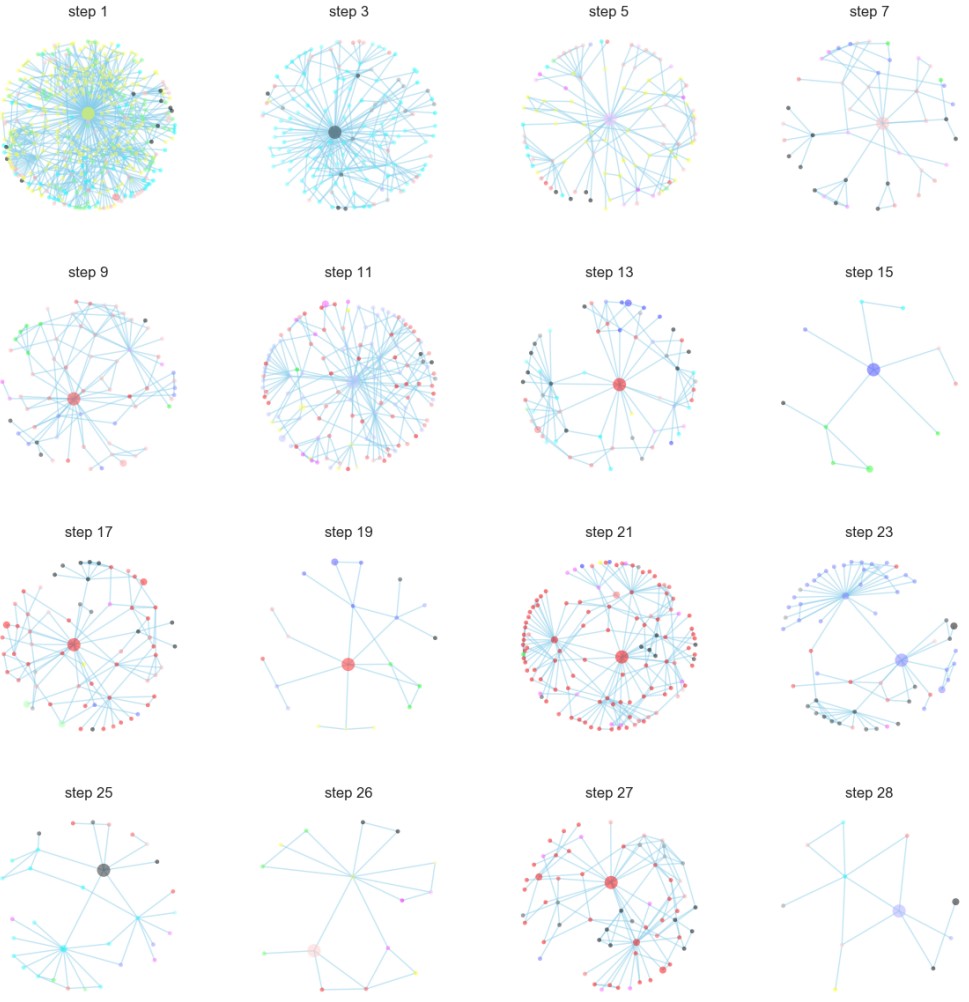

Figure 3: Case study carried out on Cora dataset

## A.2 ABLATION STUDY

We conduct contrast experiments to study the importance of different features to the model. We take DAG-Single on Cora dataset as an example. To study the importance of structure feature to the model (denoted by *-structure* in Table 5), we substitute the Struc2vec features with random vectors and report the performance of model trained with the absence of Struc2vec features. Similarly, other features are replace by zero or random vectors in order to demonstrate the their importance. From the table, we see that heuristic features and structure features both have significant benefits to the model. However, the historical features do not have positive influence on the model, possibly because the selected nodes are actually a very small fraction of selection pool, thus may not providing much information. Future study will be carried out to better leverage the historical information.

Table 5: The importance of different features in our algorithm

|  | - historical | - heuristic | - structure | DAG-Single |
|---|---|---|---|---|
| Micro-f1 | $73.2 \pm 2.4$ | $70.7 \pm 3.1$ | $70.5 \pm 2.1$ | $73.1 \pm 2.5$ |
| Macro-f1 | $66.8 \pm 5.0$ | $62.9 \pm 5.4$ | $65.9 \pm 3.5$ | $65.9 \pm 5.0$ |

