# OpenReview forum: "Transfer Active Learning For Graph Neural Networks"
_ICLR.cc/2020/Conference — Reject_

### Official Review · AnonReviewer3 · 2019-10-21
**Official Blind Review #3**

**Rating:** 3

**Review:**

In this paper, the authors proposed a new method for active learning on node classification with GCN. RL based framework is used. The labeled graph is treated as state and the action is labeling the nodes. Validation accuracy on the hold out set is used as reward. Further transfer learning framework is also proposed, where graph-specific policy and master policy are jointly learned. Experiments on benchmark dataset show the effectiveness of proposed method compared to several baselines.

The idea of applying RL on active learning with GCN seems to be new and it sounds natural and technically. Also the idea of transferring the learned policy to new graphs make sense for similar graphs. However, the empirical results are a bit weak and not convincing enough for me. Please find the detailed comments below.

1. Is the state defined on node level or graph level? Eq (1) is defined on node level, but I believe it should be a global policy on graph.
2. All the results have a rather high variance. To compare such results, the authors should make a significant test. Otherwise, one cannot say that the performance from one method is better than the other. Especially, for Table 3 and 4, DAG-distill performs not better than DAG-Joint.
3. What do "0-4, 5-9,..." mean in Table 3?
4. Can the authors show curve as in figure 2 for table 2 and 3. It is important to see the progress for active learning.
5. Why the results differ so much in Figure 2 for only 1 query? I believe the first one should be randomly picked. Thus all the methods should perform equally.
6. "Graphs encode the relations between different objects and are ubiquitous in real-world." Typo in first sentence.
7. homologous or homogenous?

**Experience Assessment:**

I have published one or two papers in this area.

**Review Assessment: Checking Correctness Of Derivations And Theory:**

I assessed the sensibility of the derivations and theory.

**Review Assessment: Checking Correctness Of Experiments:**

I assessed the sensibility of the experiments.

**Review Assessment: Thoroughness In Paper Reading:**

I read the paper at least twice and used my best judgement in assessing the paper.

---

> ### Author Response · Authors · 2019-11-13
> **Response to Reviewer #3**
>
> Thank you for your appreciation of our work and the valuable suggestions!
> 1.	As you mentioned, the state in our framework is defined on the graph level. Based on the state (i.e., the current graph), we learn a set of representations for nodes, and further compute the probability of each action (i.e., a node)
> 2.	It is a very good point to do some significance test. We will try to improve the results of our method, and also conduct significant tests in the future.
> 3.	There are 20 graphs in the PPI datasets. To better validate the effectiveness of our method, we split the 20 graphs into 4 groups (i.e., 0th – 4th graphs in the first group, 5th-9th graphs in the second group...), and we evaluate our method on each group.
> 4.	We will add the figure of the curve in our Appendix.

---

### Official Review · AnonReviewer1 · 2019-10-22
**Official Blind Review #1**

**Rating:** 3

**Review:**

Thank you for the author response.
Original review:

This paper presents a method for active learning on graphs, including a novel setting of transferring an active learning policy to unseen graphs.  The problems tackled here are important and the method is shown to improve over previous work in some cases.  On the down side, the evaluation may be missing one important method of comparison, the reasons for the proposed approach winning over previous work are not made explicit, and the empirical advantage of the approach is inconsistent (by my count, in the majority of cases the F1 advantage over previous work is within the standard error).  This paper has strengths but I feel it needs further refinement before publication.

The introduction of the paper claims that existing approaches for active learning on graphs are domain-specific and may not apply well to new domains.  But later, in the experiments, different reasons for the proposed approach's wins are given (in particular, on large graphs the proposed approach does relatively better vs. AGE which the paper suggests is due to the more complex nonlinear models used in the proposed approach).  In general, this paper’s approach tends to win over the primary baseline (the AGE method from Cai et al. 2017), but the wins are relatively small and inconsistent (esp. taking into account the standard error) whether the methods are evaluated in the single-graph setting or the transfer learning setting (of the homologous or heterogeneous variety).  If the limitation of previous work was domain-specificity, I would expect to see much larger wins on the transfer learning setting.  In general, an analysis that explains more what is driving the gains of this approach over the AGE approach would help us know how to build on this paper’s method in future work.

I was curious why the paper does not compare against the following work, which also presents an approach that wins over AGE:
“Active Discriminative Network Representation Learning,” Gao et al., IJCAI 2018

Lastly, the distillation-based approach, which learns graph-specific policies that are trained to fit their target graphs and to minimize their KL divergence from a single global shared policy, was interesting.  The fact that it doesn’t work much better than the joint policy is somewhat disappointing, but it’s still interesting.

Minor:
Sec 3.2: unclosed parenthesis in first paragraph
“Moreover, we also average the struc2vec features of all previously annotated nodes to capture the historical information” -- since the model has only node-level features, I didn’t understand how this average across multiple nodes was fed in as a node feature.  Is it used in all nodes?  Only annotated nodes?

**Experience Assessment:**

I have read many papers in this area.

**Review Assessment: Checking Correctness Of Derivations And Theory:**

I assessed the sensibility of the derivations and theory.

**Review Assessment: Checking Correctness Of Experiments:**

I assessed the sensibility of the experiments.

**Review Assessment: Thoroughness In Paper Reading:**

I read the paper thoroughly.

---

> ### Author Response · Authors · 2019-11-13
> **Response to Reviewer #1**
>
> Thank you for your appreciation and the valuable comments!
> 1.	We will try to further improve the results of our method in the paper.
> 2.	When working on this project, we didn’t notice the IJCAI paper you mentioned. Currently, we find it hard to run the method on our datasets, as the codes of the methods are not provided online. We will try to compare with the work in the future.
> 3.	Since the number of training point is so limited in the active learning setting, Under different kinds of initialization, the accuracy fluctuate dramatically. But since the reported average is calculated from 50 experiments with different random seeds, the mean accuracy is stable, so we can still argue that our method is superior to AGE.
> 4.	The possible reason for the good performance of DAG-Joint is that the two training graphs (Cora and Citeseer) are similar. In the future, we will try to conduct experiments on more diverse graphs to validate the effectiveness of DAG-Distill.

---

### Official Review · AnonReviewer2 · 2019-10-24
**Official Blind Review #2**

**Rating:** 3

**Review:**

Positive
1. The paper studies a universal policy for labeling nodes on graphs with multiple training graphs which can be transferred to new unseen graphs.
2. The paper focuses on minimizing human efforts in obtaining labeled data. The model is based on active learning and transfer learning.

Negative
1. The proposed method combines existing models as the solution, which is heuristic and lacks persuasive theoretical proofs.
2. In graphs, data (nodes) to be labeled are highly correlated. However, there is no method for solving this challenge.
3. In Section 3.2, ACTIVEL EARNING ON A SINGLE GRAPH, the authors formalize the problem, i.e., learning a policy for selecting a set of nodes for annotation, as a sequential decision process, and the reinforce algorithm is applied to optimize the objective function. However, explanation about the active learning is confusing. More details are needed to explain their respective goals and to explain how to integrate active learning and reinforcement learning.
4. The authors claim that the details of heuristic features are represented in Appendix, please add these information.
5. The settings of active learning need more consideration. The total budget for active learning is set as $5\times N_{class}$. How to choose these nodes? Is it to select all samples at once or in batches during iterative epoch? If the samples are selected in batches, what is the specific experimental setting?
6. The experimental method about active learning. The paper focuses on minimizing human efforts in obtaining labeled data. Compared to the results of the model before selecting, how significant is the improvement after selecting all the node in the budget? More experiments are needed.
7. Minor format issue: the fonts in Table 2 and Table 3 are different.


**Experience Assessment:**

I have published one or two papers in this area.

**Review Assessment: Checking Correctness Of Derivations And Theory:**

I assessed the sensibility of the derivations and theory.

**Review Assessment: Checking Correctness Of Experiments:**

I assessed the sensibility of the experiments.

**Review Assessment: Thoroughness In Paper Reading:**

I made a quick assessment of this paper.

---

> ### Author Response · Authors · 2019-11-13
> **Response to Reviewer #2**
>
> Thank you for the insightful comments!
> 1.	Active learning for graph neural networks is a challenging problem, which requires analyzing the informativeness of each node, and hence it is quite important to incorporate all different kinds of heuristics. The main focus of the paper is not on the theoretical analysis. Instead, we aim at designing a method to integrate all these heuristics for prediction. Nevertheless, deriving theoretical proofs is still an important suggestion, and we will leave it as a future work.
> 2.	Considering the correlation of data is a very intuitive suggestion. Indeed, the correlation of data is considered in our approach. This is because we leverage a GNN in our policy network, which is able to aggregate information from the neighbors of each node, and thereby capture the dependency of different nodes.
> 3.	We will try to improve the writing and make it clearer in the revised draft.
> 4.	We have added the details of the heuristic features in the updated draft.
> 5.	In our paper, we formalize the problem as a sequential decision process, therefore we select different nodes step-by-step, instead of selecting all of them at once. Sorry about the confusion in writing, and we will keep polishing the writing.
> 6.	I guess that you are wondering the performance gain of using N_seed + N_budget labeled nodes for training over using N_seed labeled nodes for training. The results are presented in the figure 3. The accuracy of training with only the N_seed nodes is lower than the starting point of the curve corresponding to “random”.
> 7.	Thank you for pointing it out! We have fixed them in the revised draft.

---

### Decision · Program_Chairs · 2019-12-19

**Decision:**

Reject

**Comment:**

Paper proposes a method for active learning on graphs. Reviewers found the presentation of the method confusing and somewhat lacking novelty in light of existing works (some of which were not compared to). After the rebuttal and revisions, reviewers minds were not changed from rejection.